# Concept of an In-Plane Displacement Sensor Based on Grating Interferometry with a Stepwise Change of Sensitivity

**DOI:** 10.3390/s21144894

**Published:** 2021-07-18

**Authors:** Leszek Sałbut, Sergiusz Łuczak

**Affiliations:** Warsaw University of Technology, Faculty of Mechatronics, 02-525 Warsaw, Poland; leszek.salbut@pw.edu.pl

**Keywords:** Grating Interferometry, moiré interferometry, in-plane displacement, strain, spatial frequency

## Abstract

Grating Interferometry, known in the relevant literature as the High Sensitivity Moiré Interferometry, is a method for in-plane displacement and strain measurement. The sensitivity of this method depends on the spatial frequency of the diffraction grating attached to the object under test. For typical specimen grating, with high spatial frequency of 1200 lines per mm, the basic sensitivity is 0.417 µm per fringe. A concept of in-plane displacement sensor based on Grating Interferometry with a stepwise change in sensitivity is presented. It is realized by using the specimen grating with lower spatial frequency. In this case, the grating has more higher diffraction orders and by selecting them appropriately, the sensitivity (chosen from 1.25 μm, 0.625 μm, or 0.417 μm) and the resulting measurement range (chosen from about 600 μm, 300 μm, or 200 μm) can be adjusted to the requirements of a given experiment. A special method of filtration is required in this case. Achromatic configuration with illumination grating was chosen due to its low sensitivity to vibration.

## 1. Introduction

It seems that as far as micro-measurements are concerned, two types of sensors are the most strategic: first, microsensors (mainly MEMS, MOEMS, and NEMS) that feature miniature overall dimensions, and second, optical sensors providing the highest sensitivity of measurements and thus enabling measurements of quantities of very low magnitudes. Such features open, for these sensors, a broad scope of sophisticated applications: both in typical areas related to engineering and medicine—as proposed e.g., in [1,2], as well as untypical, like veterinary science—as proposed e.g., in [3]. The considered sensors not only replace the typical sensors in standard applications, but also owing to their unique features are irreplaceable in completely new applications, e.g., advanced enhanced compact vehicle control systems [4]. On the other hand, even simple micro-sensors are still employed in simple applications, taking as an example detection of tilt [5].

In this paper, a concept of the second type of micro-measurements is presented: application of optical sensor for measurements of static displacements in sub-micrometer scale, employing Grating Interferometry (GI).

GI known in the relevant literature as the High Sensitivity Moiré Interferometry is a full-field optical method for in-plane displacement, strain measurement, and monitoring [6,7]. The idea of the method is presented in Figure 1. The phase diffraction grating attached to the surface of the tested object is used as the sensor of displacement. The methods of applying a diffraction grating to the sample by copying a master grating in a resin or a silicone [6,7] or employing photolithograpic techniques are well-known and described in the relevant literature [8]. This grating, called the specimen grating (SG), is symmetrically illuminated by two mutually coherent beams with plane wave fronts (Σ_A_ and Σ_B_) at the angles (*θ**_±1_*) tuned to the angle of the first and minus first diffraction order, so they satisfy the following equation:(1)$$sin\theta =\frac{\lambda }{d},$$
where: 

λ—the wavelength,

*d*—the grating period.

Then, the diffracted beams propagate along the normal to the specimen surface and interfere.

Because specimen grating deforms in the same way as the surface of the tested loaded object, the conjugated wavefronts of diffracted beams carry information about this deformation coded in the wavefronts Σ_A’_ and Σ_B’_. Intensity distribution obtained due to interference of these beams depends only on the displacements in the specimen plane, and can be described as:(2)$$I\left(x,y\right)≅1+cos\frac{4\pi }{d}u\left(x,y\right),$$
where: *u(x,y)*—the function describing in-plane displacements in a direction perpendicular to the grating lines.

The intensity distribution in the form of interference fringe pattern (see Figure 1) represents a map of in-plane displacements with the basic sensitivity equal to half of the grating period (*d*/2). For specimen grating with typical spatial frequency *f = 1/d* = 1200 lines/mm, the basic sensitivity is about 417 nm/fringe. Using the automatic fringe pattern analysis (AFPA) techniques [9], the in-plane displacements can be determined with a sensitivity of a few nanometers, and then after numerical differentiating, the map of strain can be obtained. The measurement range depends on the field of view, the local displacement gradients, and the detector resolution, and reaches the order of several micrometers (maximal value of about 200 micrometers) [6].

GI has found many applications, including a study of composite materials [10,11], fracture mechanics research [12], tests of electronics and MEMS micro-elements [13,14,15,16], analysis of residual stresses in welds [17], tests of plastics material joints [18], and many other measurement cases [6,19]. 

GI has a unique feature compared to other displacement measurement methods—the specimen grating keeps a “memory” of its initial state, which means that all successive measurements have the same reference (the moment when the grating was attached to the specimen surface), which allows one to obtain information about the cumulative displacement between the measurement periods and enables to monitor the state of the tested specimen at any time. It is a very big advantage of GI, yet on the other hand, the high sensitivity lowering the measurement range may sometimes be a problem, e.g., in the measurement of large deformations, especially their large gradients (tests of heterogeneous materials, fracture mechanics, etc.,).

In the paper, the concept of in-plane displacement sensor based on GI with a stepwise change in sensitivity is presented. It is realized by using the specimen grating with lower spatial frequency. In this case, the grating has more higher diffraction orders, and by selecting them appropriately, the sensitivity can be adjusted to the requirements of a given experiment. This also requires a special method of filtration. Achromatic configuration with illumination grating was chosen because of its low vibration sensitivity in the case of using a monochromatic light [20].

## 2. Theoretical Description

The idea of the in-plane displacement sensor with a stepwise change in sensitivity is shown in Figure 2. The beam of coherent light illuminating the phase grating IG is diffracted. The *n*-th order diffraction beams symmetrically illuminate the SG object grating. Since the IG and SG gratings have the same spatial frequencies, the diffraction angles for both gratings satisfy the equation:(3)$$sin\theta_{n}=n\frac{\lambda }{d},\;\;\;n=0,\;\pm 1,\;\pm 2,\;\ldots ,$$
where: *n*—the number of the diffraction order.

Since a value of the sine function is no higher than 1 and *n* is an integer, the value of *n* must satisfy the equation:(4)$$n\leq trunc\left(\frac{d}{\lambda }\right),$$
where: *trunc*(*x*)—a truncate function, which rounds to the nearest integer toward 0.

Note, that for 1 < *d*/*λ* < 2, the value of *n* is 1 and then a typical configuration of GI, described in the previous section, is obtained. For gratings with lower spatial frequencies (period *d* much larger than *λ*), there are more beams of higher diffraction orders illuminating the SG grating. Each beam diffracts again on SG grating and one of its diffraction orders propagates along the normal to the grating. As a result, a multi-beam interference occurs and the intensity distribution is difficult to be analyzed.

Assuming that it is possible to select and filtrate only one pair of diffraction orders (plus *n* and minus *n*), a case of two-beam interference occurs, and intensity distribution can be simply described by the following equation:(5)$$I_{n}\left(x,y\right)≅1+\cos n\frac{4\pi }{d}u\left(x,y\right),$$

Now, the interference fringes correspond to the in-displacement contour map with basic sensitivity equal to *d*/2*n*. 

Further analysis of Figure 2 shows that for points symmetrically located on the IG grating at a distance D from the optical axis, diffraction beams of the *n*-th order illuminate the SG grating at the point O_n_ located at the distance from the IG grating determined as follows:(6)$$L_{n}=\frac{D}{tan\theta_{n}}.$$

By changing the position of the SG grating from the point *O_n_* to the point *O_n+i_* and selecting appropriate diffraction orders, it is possible to change the basic sensitivity of the method. Additionally, the configuration of the sensor with two identical gratings, specimen and illuminating, is known in the relevant literature as an achromatic system with low vibration sensitivity while using a monochromatic light [20]. This enables to measure or monitor the condition of the tested object by means of the proposed sensor under non-laboratory conditions.

The key problem in the implementation of the method is to satisfy the assumption concerning the possibility of selecting diffraction orders. A novel solution to this problem is presented in the next section.

## 3. Method of Filtration of the Selected Diffraction Beams

In order to filter one selected pair of conjugated diffraction orders, a special double diaphragm F is used in the plane of the IG grating. This idea is shown in Figure 3. For the sake of simplicity, only the beams for the plus diffraction orders are shown in the figure. The diaphragm F located at a distance D from the optical axis has the diameter W. The diaphragm has been selected in such a way that the areas illuminated by the diffraction beams in the plane of the SG grating do not overlap. This condition can be written as: For diffraction orders *n* and *n* − 1:
(7)$$H_{L}=D\left(1-\frac{L_{n}}{L_{n-1}}\right)=D\left(1-\frac{tan\theta_{n-1}}{tan\theta_{n}}\right)\geq W,$$

For diffraction orders *n* and *n +* 1:

(8)$$H_{H}=D\left(\frac{tan\theta_{n+1}}{tan\theta_{n}}-1\right)\geq W,$$
where: *H_L_—*the distance between the centers of the areas illuminated by the beams of the *n* and *n −* 1 diffractive order,*H_H_—*the distance between the centers of the areas illuminated by the beams of the *n +* 1 and *n* diffractive order,*L_n_—*distance between gratings IG and SG for selected diffraction order determined according to Equation (6).

Since *H_L_* is smaller than *H_H_*, Equation (7) should be used to determine the filter diameter and the system geometry. Additionally, the central part of the diaphragm F can be removed and the interfering beams can be fed into the recording system through the window created in this way.

## 4. Concept of the Technical Implementation

The idea of the sensor system based on the proposed method is shown in Figure 4. A light beam of 671 nm generated by the diode laser (LD) is collimated by the collimating lens (CL) and split into two beams by means of the beam splitter (BS). Then, the beams are directed by mirrors (M) onto the illuminating grating (IG) with a frequency of 400 lines/mm, which is a phase type diffraction grating and according to Equation (7) it has 3 diffraction orders. The filter, described in the previous section, enables the filtration of selected diffraction orders. The selection of these orders is carried out by linear IG grating shifts along the optical axis from position I (for ±1 orders) to position II (for ±2 orders) or to position III (for ±3 orders) with distances (L1–L2) and (L1–L3), respectively (see also Figure 3). Since the SG grating period is 2.5 micrometers, the basic sensitivity of the sensor is 1.25 μm for position I (*n* = 1), 0.625 μm for position II (*n* = 2), and 0.417 μm for position III (*n* = 3).

The fringe pattern is observed and registered by the CCD matrix optically conjugated with the surface of the specimen grating. The presented system of the sensor is now under construction.

## 5. Conclusions

The concept of in-plane displacement sensor based on GI with a stepwise change in sensitivity is presented. It is to be realized using a specimen grating with a few diffraction orders. Provided they are selected correctly, the sensitivity of the sensor (1.25 μm, 0.625 μm or 0.417 μm) can be adjusted to the requirements of a given measurement by a simple linear movement of the illumination grating. Then, various measurement ranges can be obtained: about 600 μm, 300 μm, 200 μm, accordingly.

The method of filtration of selected diffraction orders is proposed and described. Achromatic configuration with illumination grating was chosen because of its low vibration sensitivity, which allows the sensor to be used under more harsh conditions than in laboratory room. Some disadvantage of the proposed method is a decrease of the light intensity for higher diffraction orders. The solution may be a special construction of a phase grating or adjusting the laser power to the selected diffraction orders.

## Figures and Tables

**Figure 1 sensors-21-04894-f001:**
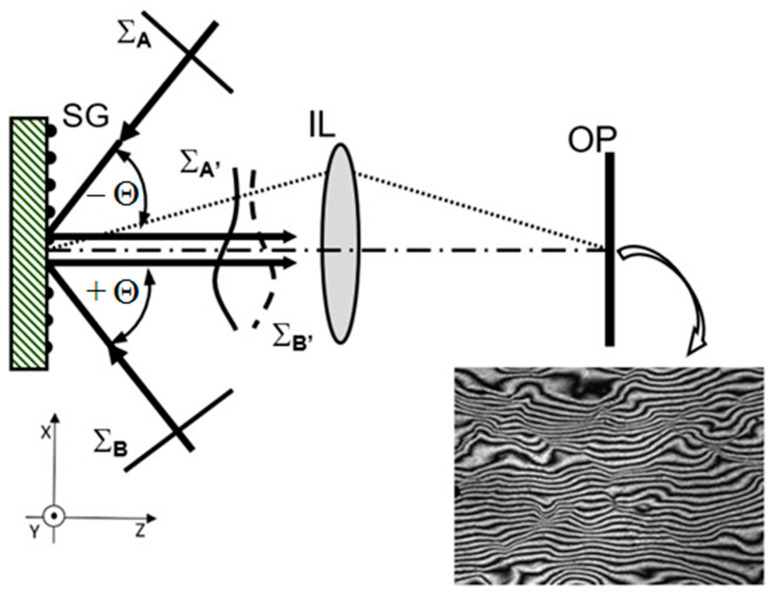
Scheme of the Grating Interferometry and an exemplary interferogram of in-plane displacements field: SG—specimen diffraction grating, *θ*_±1_—incident angles equal to the angle of plus and minus first diffraction orders, IL—imaging lens, OP—observation plane, Σ_A,B_—plane wave fronts of illuminating beams, Σ_A’,B’_—wavefronts of diffracted beams.

**Figure 2 sensors-21-04894-f002:**
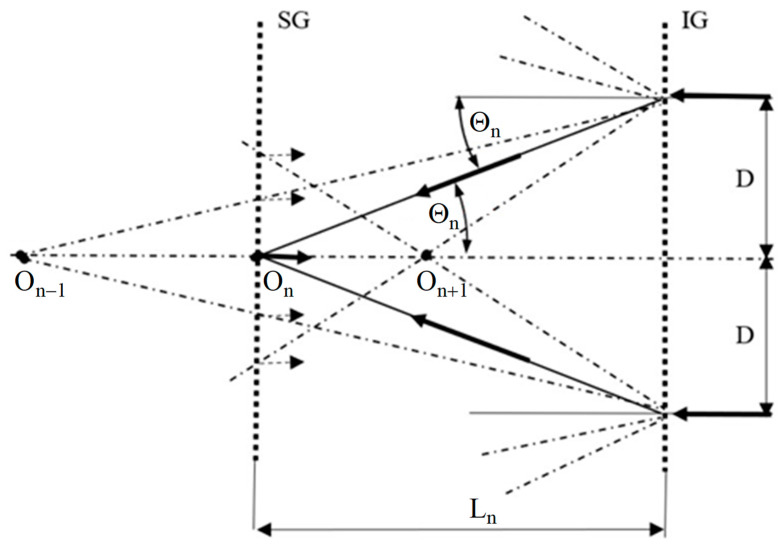
The scheme of the Grating Interferometry with a stepwise change in sensitivity: IG—illuminating grating, SG—specimen grating, *θ_n_*—angle of *n*-th diffraction order, *L_n_*—distance between the gratings.

**Figure 3 sensors-21-04894-f003:**
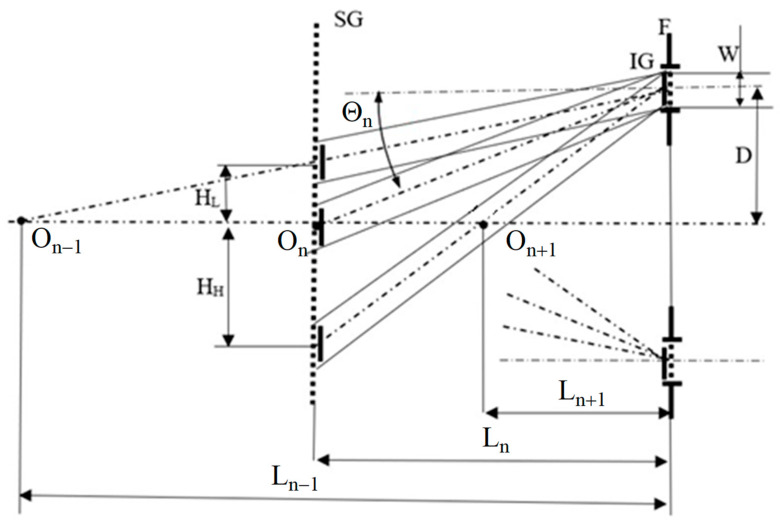
The scheme of the filtration of the selected diffraction beams: IG—illuminating grating, F—filter, *W*—filter diameter, *D—*distance between the filter and the optical axis, *H_L_*—distance between the centers of *n* and *n*-1 diffraction orders in the plane of the specimen grating SG, *H_H_*—distance between the centers of *n +* 1 and *n* diffraction orders in the plane of the specimen grating SG, *L*—distance between gratings for the selected diffraction order.

**Figure 4 sensors-21-04894-f004:**
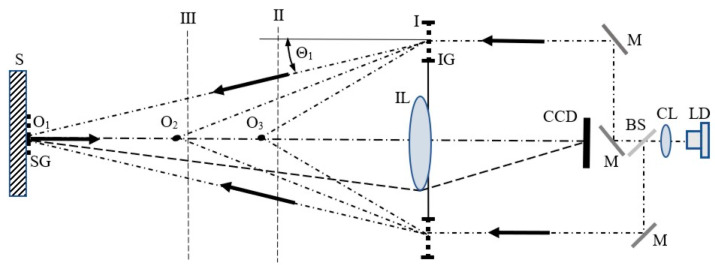
The diagram of the proposed system of in-plane displacement sensor with the gratings with frequency of 400 lines/mm: LD—laser diode, CL—collimating lens, BS—beam splitter, M—mirrors, IG—illuminating grating with filter, S—specimen, SG—specimen grating, IL—lens imaging the surface of the SG grating onto the CCD matrix.

## Data Availability

Not applicable.

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
