# Peer review of "Concept of an In-Plane Displacement Sensor Based on Grating Interferometry with a Stepwise Change of Sensitivity"

_sensors, 2021, doi:10.3390/s21144894_

Round 1

Reviewer 1 Report

The authors took into account all my previous comments and corrected errors. So, the paper has now my recommendation for publication in its present form.

Reviewer 2 Report

Dear Authors

I agree that the classification of your paper to communications helps to accept it for publication. On the other hand, it makes me more difficult to evaluate the paper's quality. Nevertheless, your effort to improve the paper is convincing, so I decided to rate the paper as acceptable.

This manuscript is a resubmission of an earlier submission. The following is a list of the peer review reports and author responses from that submission.

Round 1

Reviewer 1 Report

The paper by L. Sałbut and S. Łuczak describes a simple concept of a system based on in-plane displacement sensor with the gratings. The paper proposes to use gratings, which produce several diffraction orders, which allows adjusting the measurement range and sensitivity, depending on the used diffraction order. It is unfortunate that paper does not contain any experimental data, which is outlined as authors future work. However, I recommend the paper for publication in Sensors in its present form.

Reviewer 2 Report

The manuscript deals with a very interesting and modern topic. The fields of application are manifold: Civil construction , Conservation of artistic heritage, etc ...

However, with respect to the paper I have a series of comments-questions that are not answered in the text presented by the authors:

First, I suggest introducing the definitions of x and y reported in equation 1 in Figure 1 as well.

From a theoretical point of view, this approach certainly could work. But there are technological limitations not taken into due consideration.
First of all: what should be the resolution in mega pixels that the optical CCD sensor should have to guarantee the indicated performances?

In the text is reported that the optical source is a 671nm diode laser. How sensitive is the approach from the spectral bandwidth parameter of the source? what is the influence with respect to the coherence length of the source?

another possible problem is related to the tilt error in positioning. How do you plan to correct any imperfect parallelism between the sensor grating plane and the illuminator grating plane?

What happens if the Grating Sensor 'deforms' in a non-uniformly distributed way along the direction of the Grating itself (the pitch deforms in a non-constant way)?

Perhaps it would be better to propose this paper after a good campaign of experimental measurements

Reviewer 3 Report

It is very interesting to see how to design the GI to achieve stepwise selectable sensitivity and measurement range. I appreciate the well structured manuscript and complete introduction of the technique GI.  However, I would recommend to implement this method and demonstrate the proposed method before this manuscript being published. Besides, it would be great to clarify more about the significance in solving the problem of low measurement range.  

Reviewer 4 Report

See attached file.

Reviewer 5 Report

Dear authors

Your paper describes a displacement sensor based on grating interferometry. The novelty of your proposal is described by the diagram in fig 3 but is not clear to what extent is based on previously published work. The idea of the selection of diffraction beams is further evaluated by the diagram in fig 4 which looks realizable.  You state that the proposed sensor is under construction. Unfortunately, there are no measurement results. may I suggest that you rewrite the paper to convince the readers that the idea is working and the novel sensor is useful.